# Assessing Nutritional Status in Gastric Cancer Patients after Total versus Subtotal Gastrectomy: Cross-Sectional Study

**DOI:** 10.3390/nu16101485

**Published:** 2024-05-14

**Authors:** Fawzy Akad, Bogdan Filip, Cristina Preda, Florin Zugun-Eloae, Sorin Nicolae Peiu, Nada Akad, Dragos-Valentin Crauciuc, Ruxandra Vatavu, Liviu-Ciprian Gavril, Roxana-Florentina Sufaru, Veronica Mocanu

**Affiliations:** 1Center for Obesity BioBehavioral Experimental Research, “Grigore T. Popa” University of Medicine and Pharmacy, 700115 Iasi, Romania; fawzy_akad@umfiasi.ro; 2Department of Morpho-Functional Sciences I, “Grigore T. Popa” University of Medicine and Pharmacy, 700115 Iasi, Romania; eloae.zugun@umfiasi.ro (F.Z.-E.); dragos.crauciuc@umfiasi.ro (D.-V.C.); ruxandra-vatavu@umfiasi.ro (R.V.); liviu.gavril@umfiasi.ro (L.-C.G.); roxana-florentina.sufaru@umfiasi.ro (R.-F.S.); 3Department of Surgery I, “Grigore T. Popa” University of Medicine and Pharmacy, 700115 Iasi, Romania; 4Department of Medical Sciences II (Endocrinology), “Grigore T. Popa” University of Medicine and Pharmacy, 700115 Iasi, Romania; cristina.preda@umfiasi.ro; 5Transcend Research Center, Regional Oncology Institute, 700483 Iasi, Romania; 6Department of Vascular Surgery, “Grigore T. Popa” University of Medicine and Pharmacy, 700115 Iasi, Romania; sorin-nicolae.peiu@umfiasi.ro; 7Department of Medical Sciences II (Nephrology), “Grigore T. Popa” University of Medicine and Pharmacy, 700115 Iasi, Romania; akad_nada@d.umfiasi.ro; 8Department of Morpho-Functional Sciences II, “Grigore T. Popa” University of Medicine and Pharmacy, 700115 Iasi, Romania

**Keywords:** gastric cancer, gastrectomy, malnutrition, nutritional assessment, Nutritional Risk Screening-2002, Patient-Generated Subjective Global Assessment

## Abstract

Gastric cancer (GC) remains a significant global health concern, ranking as the third leading cause of cancer-related deaths. Malnutrition is common in GC patients and can negatively impact prognosis and quality of life. Understanding nutritional issues and their management is crucial for improving patient outcomes. This cross-sectional study included 51 GC patients who underwent curative surgery, either total or subtotal gastrectomy. Various nutritional assessments were conducted, including anthropometric measurements, laboratory tests, and scoring systems such as Eastern Cooperative Oncology Group/World Health Organization Performance Status (ECOG/WHO PS), Observer-Reported Dysphagia (ORD), Nutritional Risk Screening-2002 (NRS-2002), Patient-Generated Subjective Global Assessment (PG-SGA), and Simplified Nutritional Appetite Questionnaire (SNAQ). Serum carcinoembryonic antigen (CEA) levels were significantly higher in the subtotal gastrectomy group. Nutritional assessments indicated a higher risk of malnutrition in patients who underwent total gastrectomy, as evidenced by higher scores on ORD, NRS-2002, and PG-SGA. While total gastrectomy was associated with a higher risk of malnutrition, no single nutritional parameter emerged as a strong predictor of surgical approach. PG-SGA predominantly identified malnutrition, with its occurrence linked to demographic factors such as female gender and age exceeding 65 years.

## 1. Introduction

Gastric cancer (GC) poses a substantial global public health challenge, ranking as the third leading cause of cancer-related deaths, even though its incidence has decreased over the past five decades. The occurrence of GC exhibits regional variations, and this diversity is linked to a range of factors, including infectious, environmental, and genetic characteristics [1]. It is a tumor often linked to malnutrition and various nutritional deficiencies. The identification and effective management of these nutritional issues can play a crucial role in enhancing the quality of life and increasing the survival rates of affected patients [2]. The studies have found the prevalence of malnutrition in GC to be around 60%, although it varies widely depending on the tumor stage, type of treatment received, and the nutritional assessment (NA) tool used. Malnutrition in these patients, as with other oncological processes, results in a worse prognosis and quality of life, as well as a negative clinical impact (higher rates of infections, toxicity of treatments, complications, hospital stay, etc.) and financial impact [2,3]. The standard treatment for GC involves curative resection and perioperative adjuvant treatment. The surgical approach for GC is either total gastrectomy (TG) or subtotal gastrectomy (SG). These surgical procedures can alter the gastrointestinal anatomy and physiological function of patients, impacting nutrient absorption and giving rise to gastrointestinal symptoms [4]. Despite improvements in prognosis, over half of the patients experience recurrent disease. Identifying useful prognostic factors is crucial to enhance outcomes. The perioperative nutritional and inflammatory status influences short-term and long-term oncological outcomes, impacting complications and treatment continuation.

Following surgery, patients commonly encounter reduced appetite, early satiety with minimal food intake, or gastroesophageal reflux. They may also manifest symptoms like nausea, vomiting, bloating, abdominal discomfort, and diarrhea or encounter difficulties with swallowing, known as dysphagia [5]. Intense symptoms can contribute to heightened psychological distress and have an adverse effect on the patient’s dietary intake. Emotional distress, symptom disturbance, and inflammation associated with cancer can contribute to protein-energy wasting, resulting in a decrease in overall body weight [6]. The distinction between weight loss and cancer cachexia is essential to make, the latter being represented by loss of skeletal muscle mass and reduced energy intake. Cancer cachexia cannot be completely reversed by standard nutritional support and results in a gradual decline in physical function. Its pathophysiology involves a disturbance in protein and energy balance, typically caused by a combination of reduced food intake and abnormal metabolism [7].

Various NA tools have been developed for patients with GC, offering clinical advantages like accessibility and cost-effectiveness. However, understanding the characteristics of each tool is essential for informed clinical use [8]. Most frequently, these systems relied on objective measurements like oral energy intake, body weight, weight loss over time, subcutaneous fat loss, muscle wasting, serum protein levels, and immune competence. Subjective NA scoring systems, incorporating a variety of measurements, have been developed to enhance the sensitivity and specificity of nutritional status assessments [9]. The most used NA tools used are the Eastern Cooperative Oncology Group/World Health Organization Performance Status (ECOG/WHO PS), Observer-Reported Dysphagia (ORD), Nutritional Risk Screening-2002 (NRS-2002), Patient-Generated Subjective Global Assessment (PG-SGA) and Simplified Nutritional Appetite Questionnaire (SNAQ). While body mass index (BMI) is well-suited for population-level studies, its use can lead to inaccuracies in assessing adiposity. This is because the calculation does not differentiate between lean muscle mass and fat mass [10]. 

This research aimed to assess the frequency of postoperative malnutrition among individuals with GC who underwent either total or subtotal gastrectomy. Additionally, it examined the associations between measurable factors like laboratory results and subjective scoring systems used to evaluate nutritional status during the postoperative recovery phase.

## 2. Materials and Methods

### 2.1. Study Design and Patients 

The study was conducted prospectively and included a total of 51 patients who had undergone curative surgery for GC. Within this cohort, 22 patients underwent total gastrectomy, while 29 patients underwent subtotal gastrectomy. The enrolled patients were provided with standard nutritional supplements from the hospital throughout both the perioperative and postoperative periods. The participants were recruited from hospitalized patients at the Regional Institute of Oncology in Iasi. The patient selection was based on criteria to enhance homogeneity (age, gender, body mass index).

The project plan required a group of cooperative patients, with the diagnosis of GC being a crucial criterion for participation in the study. 

### 2.2. Serological Measurements 

Blood samples were drawn from the cubital vein for testing, which encompassed the analysis of serum protein, albumin, cell blood count, serum electrolytes Na^+^, K^+^, urea, creatinine, tumoral markers carcinoembryonic antigen (CAE), and carbohydrate antigen 19-9 (CA 19-9). Standard laboratory techniques were employed to gather the laboratory data.

### 2.3. Nutritional Assessments

The patients underwent evaluations on the following parameters: anthropometric measurements, laboratory data, the Eastern Cooperative Oncology Group/World Health Organization Performance Status (ECOG/WHO PS), Observer-Reported Dysphagia (ORD), Nutritional Risk Screening-2002 (NRS-2002), Patient-Generated Subjective Global Assessment (PG-SGA), and Simplified Nutritional Appetite Questionnaire (SNAQ).

#### 2.3.1. Eastern Cooperative Oncology Group/World Health Organization Performance Status

ECOG/WHO PS is a scale that ranges from zero (“fully active”) to three (“capable of only limited self-care”) and up to five (“dead”). Recognizing the significant insights functional status information can offer for cancer research populations, there is global interest in integrating performance measures like ECOG/WHO PS into routine follow-ups.

#### 2.3.2. Dysphagia Score

ORD was scored by the physician treating the patient. It is scored 0–4 (a score of 0 denotes no dysphagia; a score of 1: symptomatic, able to eat regular diet; a score of 2: symptomatic, altered eating/swallowing, I.V. fluids indicated < 24 h; a score of 3: symptomatic, severely altered eating/swallowing with inadequate caloric or fluid intake, I.V. fluids or FT indicated > 24 h; and a score of 4: life-threatening, due to obstruction or perforation).

#### 2.3.3. Nutritional Risk Screening-2002 

The NRS-2002 was endorsed by the European Society of Parenteral and Enteral Nutrition as the preferred screening and assessment method for hospital patients. It reflects the imbalance between the metabolic stress demand and normal nutritional intake. It includes the following components: assessment of food intake, anthropometric indicators, general health assessment, acute assessment, and assessment of involuntary weight loss. It also includes an age adjustment for patients aged over 70 years (+1). The questionnaire uses these items to calculate the overall nutritional risk score. Thus, the NRS-2002 questionnaire in GC aims to detect and manage malnutrition and nutritional deficiencies, contributing to the improvement of treatment outcomes and quality of life for patients. A score ≥3 is suggestive that the patient is at risk of malnutrition and a nutritional plan is to be followed.

#### 2.3.4. Patient-Generated Subjective Global Assessment

PG-SGA is widely acknowledged in clinical research as the standard method for evaluating the nutritional status of cancer patients, representing a modified iteration of the nutritional assessment tool, Subjective Global Assessment. The initial section of PG-SGA is filled out by the patients and is employed as a screening tool for nutritional risk/deficiency, commonly known as PG-SGA Short Form.

#### 2.3.5. Simplified Nutritional Appetite Questionnaire

SNAQ is a concise four-item instrument that includes items 1, 2, 4, and 6 from the Comprehensive Nutritional Assessment Questionnaire (CNAQ). These specific items evaluate appetite, satiety, taste of food, and the number of meals per day, respectively. The SNAQ was designed as a self-assessment screening tool, aiming for a quick and straightforward administration without the necessity for trained assessors or laboratory measurements. The total score ranges from 4 to 20. Previous validation studies propose a cutoff of ≤14 to indicate malnutrition and involuntary weight loss.

### 2.4. Statistical Analysis 

The data underwent analysis using the statistical software “Statistical Package for Social Science (SPSS)” version 20.0 for Windows. Variations between independent groups were examined through Student’s *t*-test and one-way analysis of variance. Spearman’s rank correlation coefficients were computed to assess the association between the scores and variables. Data are presented as mean ± SD. Significance was determined at *p* < 0.05. The agreement between the two assessment methods was evaluated using the Chi2 statistic. The Chi2 value ranges from 0 to 1; a value of 0.4 or less suggests that chance alone can explain the observed agreement, while a value of 1 signifies perfect concordance. Comparison between group means was performed with ANOVA. The coefficient of variation (CV%) highlights the percentage deviation between the two means, providing insights into the homogeneity of the value series. Additionally, if the examined variable comprises continuous values, the skewness test (−2 < *p* < 2) is applied to validate the normality of the value series. Using the ROC curve, the balance of specificity/sensitivity as a prognostic factor was represented.

## 3. Results

### 3.1. Demographic Characteristics 

The demographic characteristics are shown in Table 1.

Distribution by gender highlighted a higher frequency of male cases (66.7%), with a male/female ratio of 2/1 in both study groups (72.7% vs. 62.1%; *p* = 0.421). The average age in the GT group was slightly lower compared to the SG group (65.62 vs. 66.91 years; *p* = 0.728). In terms of age groups, it is noteworthy that 59.1% of patients in the TG group and 62.1% in the SG group were over 65 years old (*p* = 0.829), making 65 years a chosen threshold for subsequent comparisons. In terms of origin, it is notable that 55.3% of patients in the TG group and 53.5% in the SG group were from urban areas (*p* = 0.749).

Regarding lab work, only CEA differed significantly between the analyzed groups; in the SG group, it was significantly higher (1.85 vs. 3.97 U/mL; *p* = 0.027) (Table 2).

### 3.2. Nutritional Assessment 

A registered dietitian conducted nutritional evaluations. It is worth noting that no comparative analysis of nutritional supplements was conducted, thus hindering the identification of the most effective option.

#### 3.2.1. WHO/ECOG Score

The value series for the WHO/ECOG score was homogeneous, suggesting that statistical significance tests can be applied: variations within the range of 0–3; group mean 0.86 ± 0.92; median = 1; skewness test result *p* = 0.930.

The WHO/ECOG score of 3 (patient capable of limited self-care, spending over 50% of the time in bed or chair) was recorded in a small number of patients: 13.6% in the GT group and only 3.4% in the GS group. Approximately 80% of patients were physically active or capable of short-duration physical activities (*p* = 0.305).

Patients with an ECOG score of 2 (patients capable of self-care but struggling with relatively easy physical activities) were all over 65 years old (*p* = 0.001), with 2/3 originating from urban areas (*p* = 0.661). Those with a WHO/ECOG score of 3 (patients capable of limited self-care) were all over 65 years old (*p* = 0.001), and 3/4 were female (*p* = 0.018) (Table 3).

#### 3.2.2. Dysphagia Score

The value series for the dysphagia score was homogeneous, indicating that statistical significance tests can be applied: variations within the range of 0–2, group mean 0.51 ± 0.78, median = 0, skewness test result *p* = 1.132 (Table 3). 

A dysphagia score of 2 (capable of swallowing only semi-solid foods) was recorded in 18.2% of patients in the TG group and 17.2% in the SG group (*p* = 0.940). The TG group exhibited a more pronounced dysphagia score compared to the SG group.

Patients with a dysphagia score of 2 (capable of swallowing only semi-solid foods) were 3/4 female (*p* = 0.006), 88.9% were over 65 years old (*p* = 0.011), and 2/3 came from urban areas (*p* = 0.073) (Table 4).

#### 3.2.3. Nutritional Risk Screening-2002 

The value series for the NRS score was homogeneous, suggesting that statistical significance tests can be applied: variations within the range of 0–3; group mean 0.25 ± 0.74; median = 0; skewness test result *p* = 1.879.

An NRS score of 2 (moderate imbalance) was recorded in 9.1% of patients in the TG group and 3.4% in the SG group, while an NRS of 3 (severe imbalance) was recorded in 4.5% of patients in the TG group and 3.4% in the SG group (*p* = 0.603). A higher prevalence of malnutrition was observed in the group of patients who underwent TG.

Patients with an NRS score of 2 (moderate imbalance) were all female (*p* = 0.019), over 65 years old (*p* = 0.049), and 1/2 from urban areas (*p* = 0.725). Those with an NRS score of 3 (severe imbalance) were 1/2 female (*p* = 0.019), all over 65 years old (*p* = 0.049), and 1/2 from urban areas (*p* = 0.725) (Table 5).

#### 3.2.4. PG-SGA

The PG-SGA score values were consistent, allowing for statistical tests: ranging from 4 to 27, with a group mean of 10.84 ± 5.16 and a median of 10. The skewness test result was −1.035. The average PG-SGA score in the TG group was slightly higher than in the SG group (10.86 vs. 10.83; *p* = 0.981), emphasizing the need for dietitian intervention in both groups (Table 3).

The PG-SGA score of B, requiring dietitian intervention, was recorded in 54.5% of patients in the TG group and 62.1% in the SG group, while PG-SGA of C, a critical indicator for changing nutritional management, was recorded in 13.6% of patients in the TG group and 10.3% in the SG group (*p* = 0.856). Patients classified as severely malnourished were more prevalent in the TG group.

Patients with a PG-SGA score of B were 3/4 male (*p* = 0.026), 83.3% were over 65 years old (*p* = 0.001), and 60% came from urban areas (*p* = 0.609). Meanwhile, patients with a PG-SGA score of C were 83.3% female (*p* = 0.026), all over 65 years old (*p* = 0.001), and 2/3 came from urban areas (*p* = 0.609) (Table 6).

#### 3.2.5. Score SNAQ-Indicator of Appetite and Eating Behavior

The SNAQ score values were consistent, allowing for statistical tests: ranging from 8 to 19, with a group mean of 14.22 ± 3.29 and a median of 14. The skewness test result was −0.072 (Table 3). No patient obtained the maximum score of 20. 

The average SNAQ score in the TG group was slightly higher compared to the SG group (14.41 vs. 14.07; *p* = 0.718). A high SNAQ score was recorded in 95.5% of patients in the GT group and 93.1% in the GS group (*p* = 0.139). Surprisingly, a more satisfying appetite was found in the TG group.

Patients with a high SNAQ score were 68.8% male (*p* = 0.071), 58.3% were over 65 years old (*p* = 0.211), and 56.3% came from urban areas (*p* = 0.556). Meanwhile, patients with a low SNAQ score were all male (*p* = 0.071), over 65 years old (*p* = 0.211), and from urban areas (*p* = 0.556) (Table 7).

By plotting the ROC curve, it is confirmed that, in the studied cases, among the nutritional status parameters, there were no good predictors of radical gastrectomy (AUC < 0.600) (Figure 1, Table 8). A low AUC value near 0.5 and an irregular ROC curve suggest limited support for total gastrectomy from questionnaire results, warranting further investigation. Additionally, examining the questionnaire items and their relevance to the outcome of interest (support for total gastrectomy) may help identify any limitations or biases in the questionnaires.

## 4. Discussion

In clinical practice, a combination of these tools may be used to comprehensively assess the nutritional status of GC patients and guide personalized nutritional interventions. Early identification of nutritional issues and proactive nutritional support are essential components of the multidisciplinary care approach for GC patients, aiming to improve treatment outcomes and quality of life.

In this study, the ORD, NRS-2002, and PG-SGA scores indicated a significantly higher risk of malnutrition when TG was performed. This could be attributed to the tumor’s invasiveness that led to the TG and its local impact on nutritional assimilation. Additionally, the systemic effects of cancer cachexia may contribute to the malnourished state. There was a greater proportion of patients with a high WHO/ECOG score of 3 in the TG group. An essential feature of this study is its exclusive focus on patients who underwent gastrectomy for potentially curable gastric cancer, with the exclusion of individuals with stage IV disease. This selection criterion was applied even though cancer cachexia prevalence ranges from 50% to 80% among patients with advanced cancer [11]. 

The NRS-2002 demonstrated limited concordance with the PG-SGA regarding the diagnosis of malnutrition. NRS-2002 discovered malnutrition in only 3.9% of patients, but the PG-SGA showed different degrees of malnutrition in 70.6% (58.8% moderate and 11.8% severe). The latter method is known to be more efficient for identifying this condition. The NRS-2002 can serve as a preliminary screening tool for identifying malnutrition and the risk of malnutrition in cancer patients, preceding the utilization of the PG-SGA [12]. Risk factors associated with severe malnutrition were female sex and age over 65 years.

In alignment with our research, Jendretzki et al. similarly affirmed that women and patients aged over 65 exhibit a significantly higher prevalence of nutritional issues [13].

Malnutrition frequently occurs in critically ill cancer patients and is linked to unfavorable outcomes. This condition can be effectively addressed through nutritional interventions. Numerous investigations indicate that individuals with gastrointestinal malignancies often experience significant weight loss before surgery, with an additional approximately 10% decrease in weight observed during the initial months following the procedure [9]. Malnutrition can potentially be addressed and reversed through straightforward methods like refeeding and nutritional supplementation via oral intake, enteral tube feeding, or parenteral nutrition. However, cancer cachexia presents as a complex syndrome, representing a specific type of chronic disease-related malnutrition accompanied by inflammation. Due to the intricate interconnection of its various components, there is no singularly effective treatment available for cancer cachexia [14,15]. According to the literature, Aydin et al. reported that malnutrition may exist despite a normal BMI, and the SGA can identify malnutrition even before the BMI falls below 20 kg/m^2^. BMI serves as a straightforward tool, readily applicable in clinical settings; however, it cannot distinguish between fat and muscle mass, necessitating repeated measurements for clinical relevance. As individuals age, adiposity tends to increase while muscle mass declines, often without notable BMI alteration, therefore the concept of sarcopenic obesity warrants consideration. Hence, employing a combination of assessment methods is crucial for evaluating a patient’s nutritional status effectively [16,17]. 

Several studies have shown a correlation between low serum albumin levels and extended hospital stays, medical complications, and higher mortality rates. However, conflicting findings indicate that low serum protein levels do not consistently signify malnutrition, nor does malnutrition always coincide with low serum protein levels. Inflammatory responses, liver disease, cancer, or idiopathic factors can influence various serum proteins and albumin levels [9,14,18].

Nutritional status, food intake, and disease severity should be evaluated regularly and at short intervals, ideally at least every 1–2 months, from the initial contact to promptly identify any decline in nutritional status [19]. Elevated scores on the NRS-2002 tool have been linked to higher rates of postoperative complications and prolonged hospital stays. Additionally, in a recent investigation assessing the effectiveness of the SNAQ in predicting postoperative mortality risk following GC surgery, an SNAQ score ≥1 was associated with an increased mortality rate compared to an SNAQ score below this threshold [20,21]. The SNAQ lacks adequacy as a predictive tool for identifying nutritional risk in ambulatory cancer patients, primarily due to its failure to consider specific factors like tumor stage and adverse treatment effects [22]. 

A recent study conducted by Chen et al. evaluated four NA tools, including SNAQ, NRS-2002, Universal Screening Tool (MUST), and the Malnutrition Screening Tool (MST). The results indicated that the former three exhibited favorable sensitivity and specificity. However, among patients with gastric cancer, MST exhibited superior performance in identifying cachexia, as evidenced by its high sensitivity (84.3%), specificity (98.6%), and AUC (0.914, *p* < 0.001). On the other hand, the SNAQ tool displayed high specificity but relatively low sensitivity, thereby restricting its diagnostic effectiveness [15].

Hauner et al. (2020) conducted an assessment of MUST and NRS-2002 in outpatient cancer populations, finding both tools to be suitable, although their efficacy varied depending on the tumor type. They noted that patients with digestive tumors exhibited a higher prevalence of malnutrition according to MUST, with rates reaching 46.6%, whereas NRS-2002 identified a higher percentage of malnutrition, at 63.3%, among patients with hematopoietic tumors. Hence, it appears crucial to take into account that the tumor type also influences the selection of the most appropriate nutritional screening tool based on its specific characteristics [23].

Postoperative nutritional support plays a crucial role in preserving nutritional status during the catabolic phase following surgery. Shim et al. examined the perioperative nutritional status of 435 GC patients, showing a notable rise in the prevalence of severe malnutrition post-surgery (from 2.3% before surgery to 26.3% after surgery) [24]. Following surgical intervention, both appetite and dietary intake tend to decrease during the recovery period, with nutritional status requiring up to a year to fully recuperate [9]. Early nutrition can be safely initiated 6 h after surgery through a percutaneous jejunostomy tube. This early postoperative nutrition strategy helps mitigate the heightened metabolism associated with surgical trauma, preserves the integrity of the intestinal mucosal barrier, reduces the risk of intestinal-borne infections, and promotes the overall recovery of patients [25].

### Study Limitations

The study’s limitations include its small sample size of only 51 patients and its cross-sectional design allowing only for associations to be determined, without establishing causality between variables. For instance, the cross-sectional approach prevented the assessment of nutritional status changes over time and their impact on recovery. Furthermore, lacking data on nutritional status at cancer diagnosis hindered its consideration as a covariate in regression models. It is important to note that this study recruited a convenience sample from a single medical center, limiting the generalizability of the results to all GC patients. Confounding factors may lead to inaccurate estimates of the true association between variables. Despite these limitations, the study offers valuable insights into factors affecting the nutritional status of GC patients.

## 5. Conclusions

Patients afflicted with malignant gastrointestinal conditions frequently experience a heightened prevalence of malnutrition. In the context of cancer, diminished food intake and an increased energy deficit contribute to the decline in nutritional well-being. It holds significant importance to identify malnourished individuals both before surgery and throughout the postoperative monitoring phase. While objective nutritional parameters offer valuable insights, subjective assessments also play a role despite their inherent limitations in accurately gauging nutritional status. NRS-2002 is a viable option for initial screening before administering PG-SGA. Our study revealed that NRS-2002 showed only partial agreement with the PG-SGA in diagnosing malnutrition, while the ORD, NRS-2002, and PG-SGA scores collectively pointed to a notably increased risk of malnutrition following TG. This may be linked to the tumor’s invasiveness necessitating TG. As such, evaluating the nutritional status of patients following gastrectomy necessitates a blend of objective metrics such as anthropometric measurements and laboratory tests, alongside subjective scoring systems during the postoperative observation period. Further investigation is warranted regarding nutritional screening tools tailored for cancer patients. This emphasizes the pressing requirement for additional studies to assess and compare existing tools, alongside the development of novel ones more suited to this condition, incorporating the factors outlined in this study.

## Figures and Tables

**Figure 1 nutrients-16-01485-f001:**
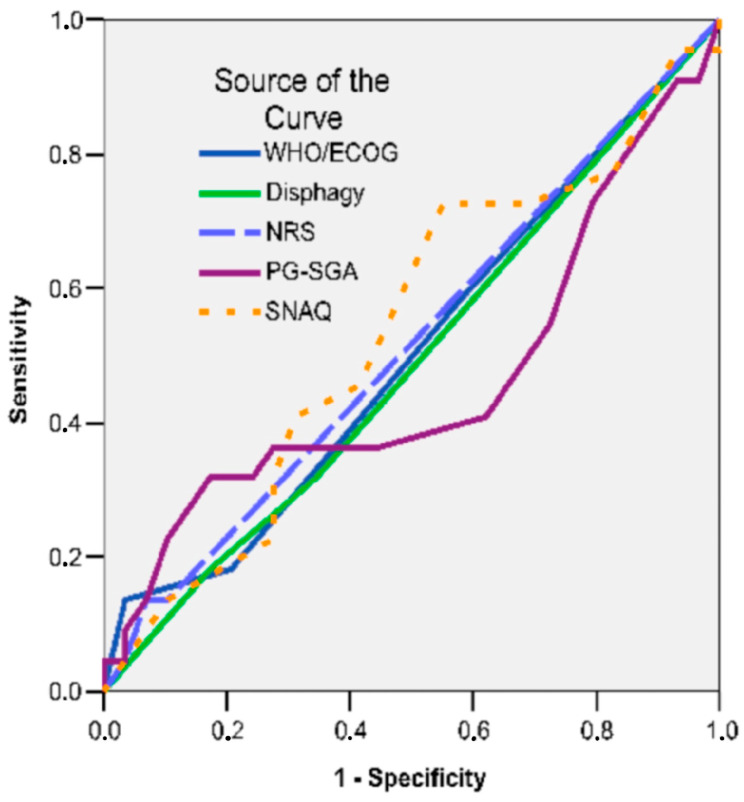
ROC Curve: Nutritional Status. Predictors of Radical Gastrectomy.

**Table 1 nutrients-16-01485-t001:** Demographic characteristics of the patients.

	Gastrectomy		Both Groups (N = 51)	*p*-Value
	Total (N = 22)	Subtotal (N = 29)		
**Sex** Male Female	16 (72.7%) 6 (27.3%)	18 (62.1%) 11 (37.9%)	34 (66.7%) 17 (33.3%)	0.421
**Age (years)** Mean ± SD median/limits	66.91 ± 12.68 66/35–82	65.62 ± 13.32 65/34–82	66.18 ± 12.93 66/34–82	0.728 ^#^
**Age groups** <65 years ≥65 years	9 (40.9%) 13 (59.1%)	11 (37.9%) 18 (62.1%)	20 (39.2%) 31 (60.8%)	0.829
**Environment** Urban Rural	13 (59.1%) 9 (40.9%)	16 (55.2%) 13 (44.8%)	29 (56.9%) 22 (43.1%)	0.779

^#^ Student’s *t*-test.

**Table 2 nutrients-16-01485-t002:** Biochemical parameters compared between the two groups.

Parameters	Gastrectomy		*p*-Value
	Total (N = 22)	Subtotal (N = 29)	
Hemoglobin, g/dL	10.43 ± 2.81	11.36 ± 2.41	0.209
Hematocrit, %	32.57 ± 7.65	34.97 ± 6.07	0.217
PLT, ×10^3^/µL	330.59 ± 131.43	337.76 ± 120.98	0.841
WBC, ×1000/µL	7.59 ± 2.04	8.11 ± 2.84	0.474
Lymphocytes/µL	9.25 ± 9.28	7.64 ± 10.32	0.567
PLT/Lymphocytes	109.13 ± 97.39	137.42 ± 93.71	0.299
Serum protein, g/dL	7.13 ± 0.71	6.96 ± 0.96	0.482
Albumin, g/dL	4.22 ± 0.63	4.40 ± 0.61	0.325
Na^+^, mmol/L	138.68 ± 3.24	139.90 ± 3.71	0.228
K^+^, mmol/L	4.53 ± 0.38	4.41 ± 0.44	0.345
Urea, mg/dL	40.0 ± 11.69	43.18 ± 15.53	0.426
Creatinine, mg/dL	0.99 ± 0.21	1.01 ± 0.23	0.783
CEA, U/mL	1.85 ± 2.27	3.97 ± 1.63	0.027
CA19-9, U/mL	53.55 ± 32.80	43.03 ± 26.69	0.803

**Table 3 nutrients-16-01485-t003:** Distribution of cases based on ECOG/WHO PS scores and demographic characteristics.

ECOG/ WHO PS		0 N = 21 (41.2%)	1 N = 20 (39.2%)	2 N = 6 (11.8%)	3 N = 4 (7.8%)
Surgical treatment	SG (N = 29)	12 (41.4%)	11 (37.9%)	5 (17.2%)	1 (3.4%)
	TG (N = 22)	9 (40.9%)	9 (40.9%)	1 (4.5%)	3 (13.6%)
Sex	Male	17 (80.9%)	13 (65%)	3 (50%)	1 (25%)
	Female	4 (19.0%)	7 (35%)	3 (50%)	3 (75%)
Age	<65 years	17 (80.9%)	4 (20%)	0	0
	≥65 years	5 (23.8%)	16 (80%)	6 (100%)	4 (100%)
Environment	Urban	10 (47.6%)	13(65%)	4 (66.6%)	2 (50%)
	Rural	11 (52.3%)	7(35%)	2 (33.3%)	2 (50%)

SG = subtotal gastrectomy; TG = total gastrectomy. 0 = asymptomatic; 1 = symptomatic but completely ambulatory; 2 = symptomatic, <50% in bed during the day; 3 = symptomatic, >50% in bed but not bedbound.

**Table 4 nutrients-16-01485-t004:** Distribution of cases based on dysphagia scores and demographic characteristics.

Dysphagia Score		0 N = 34 (66.7%)	1 N = 8 (15.7%)	2 N = 9 (17.6%)
Surgical treatment	SG (N = 29)	19 (65.5%)	5 (17.2%)	5 (17.2%)
	TG (N = 22)	15 (68.2%)	3 (13.6%)	4 (18.2%)
Sex	Male	27 (79.4%)	5 (62.5%)	2 (22.2%)
	Female	7 (20.6%)	3 (37.5%)	7 (77.7%)
Age	<65 years	18 (52.9%)	1 (12.5%)	1 (11.1%)
	≥65 years	16 (47.0%)	7 (87.5%)	8 (88.8%)
Environment	Urban	16 (47.0%)	7 (87.5%)	6 (66.6%)
	Rural	18 (52.9%)	1 (12.5%)	3 (33.3%)

SG = subtotal gastrectomy; TG = total gastrectomy. 0 = no dysphagia; 1 = symptomatic, able to eat a regular diet; 2 = symptomatic, altered eating/swallowing.

**Table 5 nutrients-16-01485-t005:** Distribution of cases based on NRS-2002 scores and demographic characteristics.

NRS-2002 Score		0 N = 44 (88.2%)	1 N = 1 (2%)	2 N = 3 (5.9%)	3 N = 2 (3.9%)
Surgical treatment	SG (N = 29)	26 (89.7%)	1 (3.4%)	1 (3.4%)	1 (3.4%)
	TG (N = 22)	19 86.4%	0	2 (9.1%)	1 (4.5%)
Sex	Male	33 (73.3%)	0	0	1 (50%)
	Female	12 (26.7%)	1 (100%)	3 (100%)	1 (50%)
Age	<65 years	20 (44.4%)	0	0	0
	≥65 years	25 (55.5%)	1 (100%)	3 (100%)	1 (100%)
Environment	Urban	25 (55.5%)	1 (100%)	2 (66.6%)	1 (50%)
	Rural	20 (44.4%)	0	1 (33.3%)	1 (50%)

SG = subtotal gastrectomy; TG = total gastrectomy. Score < 3—no nutritional risk; >3—nutritional risk.

**Table 6 nutrients-16-01485-t006:** Distribution of cases based on PG-SGA scores and demographic characteristics.

PG-SGA Score		A N = 1 (29.4%)	B N = 30 (58.8%)	C N = 6 (11.8%)
Surgical treatment	SG (N = 29)	8 (27.6%)	18 (62.1%)	3 (10.3%)
	TG (N = 22)	7 (31.8%)	12 (54.5%)	3 (13.6%
Sex	Male	11 (73.3%)	22 (73.3%)	1 (16.7%)
	Female	4 (26.7%)	8 (26.7%)	5 (83.3%)
Age	<65 years	15 (100%)	5 (16.7%)	0
	≥65 years	0	25 (83.3%)	6 (100%)
Environment	Urban	7 (46.7%)	18 (60%)	4 (66.6%)
	Rural	8 (53.3%)	12 (40%)	2 (33.3%)

SG = subtotal gastrectomy; TG = total gastrectomy; A = well nourished; B = suspected or moderate malnutrition; C = severe malnutrition.

**Table 7 nutrients-16-01485-t007:** Distribution of cases based on SNAQ scores and demographic characteristics.

SNAQ Score		Low N = 1 (2%)	Moderate N = 2 (3.9%)	High N = 48 (94.1%)
Surgical treatment	SG (N = 29)	0	2 (6.9%)	27 (93.1%)
	TG (N = 22)	1 (31.8%)	0	21 (95.5%)
Sex	Male	1 (100%)	0	33 (68.8%)
	Female	0	2 (100%)	15 (31.3%)
Age	<65 years	0	0	20 (41.7%)
	≥65 years	1 (100%)	2 (100%)	28 (58.3%)
Environment	Urban	1 (100%)	1 (50%)	27 (56.3%)
	Rural	0	1 (50%)	21 (43.7%)

SG = subtotal gastrectomy; TG = total gastrectomy. Low = low risk of malnutrition; Moderate = moderate risk of malnutrition; High = high risk of malnutrition.

**Table 8 nutrients-16-01485-t008:** ROC curve and AUC for the studied tests.

Test Result Variable (s)	Area	Std. Error (a)	Asymptotic Sig. (b)	Asymptotic 95% Confidence Interval	
				Lower Bound	Upper Bound
WHO/ECOG	0.505	0.083	0.947	0.343	0.668
Dysphagia	0.491	0.082	0.909	0.329	0.652
NRS-2002	0.518	0.083	0.827	0.356	0.680
PG-SGA	0.464	0.087	0.662	0.294	0.634
SNAQ	0.536	0.083	0.662	0.373	0.699

The test result variable(s), WHO/ECOG, dysphagia, NRS-2002, PG-SGA, SNAQ, has at least one tie between the positive and negative actual state groups. Statistics may be biased. a: under the nonparametric assumption. b: null hypothesis; true area = 0.5.

## Data Availability

The patient database can be found in the Regional Oncology Institute, 700483 Iasi, Romania.

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
