# Peer review of "Assessing Nutritional Status in Gastric Cancer Patients after Total versus Subtotal Gastrectomy: Cross-Sectional Study"

_nutrients, 2024, doi:10.3390/nu16101485_

Round 1
Reviewer 1 Report
Comments and Suggestions for Authors
Veronica et al. submitted the manuscript entitled: Assessing Nutritional Status in Gastric Cancer Patients after total versus subtotal gastrectomy: cross-sectional study, in which they specially focused on GC patients after TG surgery and compared among the applicability among different questionnaires. In general, this topic will be of interest to the potential readers of Nutrients. However, as a cross-sectional study, the authors should include more discussions on comparison of questionnaires.
My comments are as follows:
1. Discussion section: Here I suggested some aspects for further discussion and comparison. The authors can include aspects such as primary focuses, scopes of assessment and usage in clinical practice. In the same time, the authors can generally discuss on similarities and differences for GC patients before and after TG surgery and elucidate how these factors influence on selection of questionnaires.
2. Page 3, section 2.1: Did the recruited patients receive any nutrition supplement in perioperative and postoperative period? If yes, did all of them receive the same standard nutrition supplement?
3. Based on the questionnaires and results, are the authors able to identify which kinds of nutrients are essential for GC patients after TG?
Author Response
- We have added a paragraph about the clinical practice of NA- page 8, section 4; The aim of the research is stated on page 2, section 1, line 92; We did not assess the patients before the surgery, thereby precluding any comparative analysis in that regard. We have added that status of the nutritional supplementation- page 3, section 2.1, line 101. The patients did not receive a certain diet. The study did not focus on various nutritional supplementations – page 4, section 3.2, line 192. The conclusions have been enhanced with factual information derived from both the study findings and existing literature- page 11, section 5.
Reviewer 2 Report
Comments and Suggestions for Authors
1. ROC figure for predicting surgery is not much meaningful as it has a lot of confounding factor will affect the patients to do or not do radical gastrectomy.
Author Response
- We have provided a clearer explanation of the ROC figure to prevent misinterpretation- page 7, section 3, line 274.
Reviewer 3 Report
Comments and Suggestions for Authors
Akad et al present a cross-sectional study on a hot medical issue. The methodology adopted has been thoughtful and been presented properly. The results are adequately presented. Their interpretation is correct. No plagiarism or language flaws have been detected.
Minor comments:
More literature references might support their results and authors are advised to add more references.
They have to write more clearly the limitations of their study and its value.
Author Response
- A few more references relevant to the topic have been incorporated. We have revised and clarified the study limitations for better clarity.
Round 2
Reviewer 1 Report
Comments and Suggestions for Authors
All raised issues have been well addressed.